# Investigating Consumer Values of Secondhand Fashion Consumption in the Mass Market vs. Luxury Market: A Text-Mining Approach

**H M Rakib ul Hasan \*, Chunmin Lang and Sibei Xia**

Department of Textiles, Apparel Design and Merchandising, Louisiana State University, Baton Rouge, LA 70803, USA
* Correspondence: hhasan3@lsu.edu

**Abstract:** The purpose of this research is to investigate consumer values of secondhand fashion (SHF) consumption from online platforms in both the mass market and luxury market. The luxury SHF business is closely related to the business of SHF mass market. A cross-market value analysis would provide better insights to understand consumers' motivations to purchase secondhand fashion products. Three mass-market SHF companies and three luxury SHF companies were selected as sample companies based on their revenues in the U.S. market. Consumers' comments and reviews from a third-party business review site, trustpilot.com, were collected using the web-scraping technique. Recurrent Neural Network (RNN) analysis, as part of a machine learning algorithm, was applied to detect the most co-occurring word combinations and underlying values discussed in the reviews. The findings identified major consumer-perceived 'source of values', i.e., 'possession/ownership transfer', and 'interaction between consumer to business platform', which might negatively impact the performance of the business of SHF mass market compared to the luxury SHF business. The 'possession/ownership transfer' source-related values are the most ignored value in the SHF mass market. By adopting the product-oriented value strategy practiced in the luxury market, SHF mass market might improve the consumer perception of product-related value areas.

**Keywords:** secondhand fashion; mass market; luxury market; consumer value; text-mining

## 1. Introduction

### 1.1. Research Background

The fashion industry is generating an unprecedented amount of waste followed by landfill causing environmental pollution. Every second, the fashion waste equivalent to one garbage truck is burned or landfilled [1]. Therefore, the recyclable value of the material is getting lost. An estimation shows that up to 95% of clothing could be recyclable [2]. However, the appropriate disposal procedure for post-consumption fashion waste is often ignored. There is no regulation enforced to protect the environment from pollution from post-consumption fashion waste. According to the Resource Conservation and Recovery Act (RCRA) Regulations, fashion products, including clothing and footwear, are considered non-durable solid waste, and their disposal is not well regulated like hazardous material disposal (e.g., car battery, tire, motor oil, etc.) [3].

In the United States, discarded fashion items are the major source of textiles in municipal solid waste (MSW). Since 2000, discarded clothing and footwear waste have doubled in 2018. Among these, the recycling rate has never been above 15% [3]. In 2018, the textile MSW was 13 million tons (4.4% of total MSW) with recycling of 1.7 million tons (13.01%). In the U.S., compared to plastic bottles, glass, and paper, with recycling rates of 29%, 27%, and 66%, respectively, it is understood that fashion recycling lags [4]. Only 12% of the material used for fashion products is recycled globally [4]. Since 2015, clothing and footwear have become the single largest source of non-durable MSW [3]. Fashion products can be recycled

that turned into new fashion products. For example, old woolen jumpers can be converted into carpets [4].

Post-consumption waste is being generated at an unprecedented speed. Without an appropriate treatment of post-consumption materials, preventive measures would impact little on the environment. One could target the re-circulation of the product to the supply chain through Collaborative Consumption (CC) practice, which extends the product lifespan. The definition of a pre-used or secondhand fashion (SHF) product has evolved from a functional and economical commodity that meet the fashion demands of the low-income community to a commodity that is pursued by customers of all classes [5]. SHF is no more purchased just because of its low price or cost-effectiveness [6]. Hedonic reasons are another major motivator of secondhand product usage [7]. Shopping for SHF product is also connected with the pleasure of finding low-cost and high-value items because SHF product allows consumers to seek unique products at an affordable price [8]. In order to express individuality, people with a strong demand for originality were found to be far more likely to make unconventional choices, for example, choosing secondhand stores over typical ones when buying clothing [7,9]. Although the SHF product has already lost some parts of its utility, the product life is not fully exhausted. If a consumer of SHF products is satisfied with the consumption, he or she may adopt the consumption of SHF products regularly and recommend this consumption activity to his or her peers. This might help the SHF business to achieve economic sustainability in the long run.

### 1.2. Research Rationale and Objectives

Nowadays, Consumers are actively looking for fellow consumers' online opinions and reviews about their products and consumption experiences [10]. Understanding a consumer's purchasing behavior, based on the entire purchase procedure, needs to establish the link from attitude to purchase through purchase intention [11]. The performance of SHF products and the possibility of sustaining the SHF consumption practice among consumers should be better understood from real consumers' post-purchase behavior. Positive word-of-mouth and repeat purchase might be important indicators for a business to run successfully. Therefore, the post-purchase stage of fashion consumption should be helpful to understand the consumers' value perception towards the product and service of the SHF resale industry, which is largely ignored in extant literature.

The types of products and business conditions are likely to influence how consumers perceive the values of products and businesses. SHF business is mainly available for two different types of SHF products: mass-market secondhand and luxury secondhand. Mass-market fashion is responsible for the major portion of material used and discarded as landfill in municipal solid waste (MSW) than luxury secondhand fashion. On the other hand, due to its hedonic value and high-cost of the luxury product, consumers might not be interested to throw it into a landfill. Therefore, mass-market SHF needs the foremost attention. In regard to the format of the secondhand fashion marketplace, the offline SHF market is mainly available in the Business to Consumer (B2C) format. On the other hand, the Online SHF market is mainly available in C2C format [10]. There are some major online C2C platforms for SHF sales namely eBay, Facebook, Gumtree, Depop, Amazon, etc. [12]. However, the online format did not get enough attention from the researchers.

In the online secondhand fashion business, typical offline SHF motivations like social-familiar relationships might be absent as there is no physical visit to the store, and consumers do not see the seller. However, some other motivations might be valid in an online environment as well as at the physical stores, for instance, the pleasure of treasure hunting, and the enjoyment of finding some items at a lower price that they may not afford otherwise. Besides, there may be some online C2C platform-specific values that have not been discussed in SHF literature. This current study will mainly focus on the online secondhand fashion marketplace.

In major online consumer review platforms, overall negative sentiment has been observed among the mass-market SHF buyers towards the product and services provided

by the SHF mass market [13,14]. The luxury SHF market shows the opposite [13,14], which can provide a guidance for SHF mass market. The formats are similar in terms of selling secondhand fashion products. However, these show the opposite perception from consumers. A comparative analysis between these two formats might put insights on how to improve consumer sentiment in the SHF mass market.

There are some variations in the drivers of value perception between mass fashion and luxury fashion brands in terms of products, services, and perceived risks [15]. For fashion resale, these values provided by brands are replaced by values associated with resale platforms. Therefore, these variations should be noticeable in luxury and mass secondhand fashion resale platforms. Undervalued markets could search for potential value from other related businesses and adopt these values to their products and services. The purchase of luxury products usually contains more hedonic-associated values, which may not be transferrable to the mass market. However, there are service-related values in the resale of luxury fashion products, which might be adopted in the mass market. Therefore, studying consumers' perceived values of secondhand fashion in both the mass market and the luxury market may provide more insights to better understand consumers' motivations and concerns, and will also provide a guidance for SHF mass market to improve business values and consumer satisfaction. A cross-market analysis among the available related formats of SHF should produce valuable implications for business managers and the resale industry. Therefore, the research aims to investigate (i) how consumers perceive the values created by the online C2C SHF business platforms and (ii) what kind of value strategies can be adopted into the mass market of SHF from luxury SHF business platforms.

## 2. Literature Review

### 2.1. Collaborative Consumption and Sustainability

The collaborative consumption activities have been grouped into three systems [16]: (1) Collaborative lifestyles system: similar-mentality people share or exchange intangible things, e.g., money, skills, space, and time. (2) Redistribution markets system: A system where the used product is transferred to a place that needs from another place that no longer needs, e.g., secondhand selling platforms, bartering, etc. (3) Product service system (PSS): a company serves goods in the form of service but does not sell it as a product. The system enables the renting or sharing of individually owned products. It changes the consumer mindset to use without owning [16,17]. The last two systems are based on product consumption. This consumption involves two different modes of exchange; one is the transfer of ownership, and the other is accessing to the usage without ownership [18,19]. Based on the transfer of ownership and usage, the system is further divided into five categories namely redistribution, disposition after the temporary acquisition, long-term access, short-term access, and mutualization [20].

The redistribution market system is the focus of this study. The term 'redistribution' is the most regular form of the product disposal system, whereby consumers dispose of products by swapping, bartering, legacies, secondhand marketplaces, or donations [9,21,22]. Individuals get together on an online platform or at a flea market to dispose of their used products in the form of private selling and buying [23]. Therefore, the system involves the permanent transfer of ownership, and monetary compensation or non-monetary compensation. In this system, consumers are encouraged to reuse and resell used items rather than send those to the waste bin. It keeps the product circulating, maximizes its usage, and extends the longevity of the product, thus, reducing waste and new production [16].

Secondhand fashion consumption is a part of the broad idea of collaborative consumption. The secondhand fashion market brings benefits from different perspectives. The usage of secondhand fashion products maximizes the lifespan of those items and reduces landfill waste [16]. The availability of secondhand fashion products will also reduce consumers' need for products made with new materials, which decrease the carbon footprint needed to produce new products in the end. Consumers also benefit from the usage of secondhand fashion. The availability of pre-owned fashion items provides individuals access to more

fashion products with lower financial input, and they are also able to obtain some fashion products that may not be affordable otherwise [24].

### 2.2. Secondhand Fashion: Mass and Luxury Market

There are two online channels for secondhand fashion products. One is the mass market and the other is the luxury consignment [25]. The term 'Mass-market secondhand fashion' is not commonly cited; however, it should be any secondhand fashion products that not fall in the category of luxury fashion. Due to the volatile nature of fashion trends [26], fashion products outdate quickly even in good condition, resulting in the donation of those products to secondhand stores [27]. This helps to create the mass-market of secondhand fashion. In the virtual world, the model works in C2C format.

In recent years, the SHF industry has received increased attention. It has formed a niche market in the western world [28] and is also a growing market all around the world. Fashion leaders in developed countries have recognized SHF as a trend. On the other hand, it is an affordable fashion alternative for consumers who live below the poverty line in underdeveloped countries [29]. Globally, the resale of secondhand fashion exists in more than 100 countries [30]. The approximate amount of global trade in the secondhand apparel market was USD 11 billion in 2012 [31]. The market was a USD 38 billion industry, and it is expected that the growth will reach USD 64 billion by 2024 [31]. Every year, roughly 12–15 percent of Americans purchase at resale or consignment stores [32]. Therefore, the secondhand fashion industry is growing. SHF mass market should be able to re-circulate the major volume of fashion products more than the SHF luxury market. In near future, SHF mass market might capture a significant portion of the global fashion supply.

The luxury secondhand fashion market is a closely related model to the mass market of secondhand fashion in terms of product exchange system. Both need a platform to facilitate the engagement of firsthand and secondhand consumers. In terms of consumer-to-consumer interaction, both types of platforms may allow a direct consumer-to-consumer interaction (C2C) or work purely as a mediator. The latter is known as the consignment business model. The consignment platform authority accepts products for sale and pledges to pay a percentage of the revenues to the seller if the products are sold [33]. Some platforms may pay for the products immediately once they are received and before they are sold [34,35]. Consignment store bridges the consumer-to-consumer (C2C) product transfer and intervenes in the process more than C2C SHF platforms. Therefore, it could be termed C2B2C.

Luxury goods were one of the fastest-growing secondhand product categories in 2018 [36]. The approximate amount of global revenue of the secondhand luxury market is USD 4.9 billion in 2021 and is projected to grow up to USD 14.6 billion by the year 2027. The rapid growth of online networks–such as C2C marketplaces, mobile apps dedicated to secondhand, and specialized platforms–has given pre-owned luxury goods worldwide visibility [37]. Shifting expectations and renewed acceptance of wearing and buying previously used items are influencing the growth of the secondhand luxury industry [38,39]. It has also become a means of locating rare and exclusive objects [40,41] and gaining access to luxury products that would otherwise be out of reach financially [42].

Researchers have investigated the means and values attached to the sale of second-hand luxury fashion products [43], motivations to consume secondhand luxury fashion, comparison to first-hand luxury products [25], decision-making style [44], and so forth. Secondhand luxury consumption has its characteristics because it sits at the crossroads between thrift and luxury consumption [40]. However, as a niche market of the secondhand industry, it should have some transferable service values that could be adopted by the resale of mass-market secondhand fashion products.

### 2.3. Basic Value Framework

In the extant literature, the SHF value perspective covers basic value structure. To SHF consumers, SHF attributes show positive functional, social, and psychological consequences

(i.e., satisfaction and positive feeling) that indicates positive value perception. On the other hand, to the non-SHF consumer, SHF attributes show negative functional, social, and psychological consequences (i.e., risk perception and negative feeling) that indicate negative value perception [45]. Overall, the consequences indicate utilitarian, hedonic, symbolic, and cost-value perspectives of the consumer's perceived value structure [45]. However, the values have emerged solely from the product attributes. Moreover, SHF consumers' perceptions of negative values and non-SHF consumers' perceptions of positive values were not explored.

The antecedents of consumer attitude toward buying secondhand fashion products are followed by the intention to purchase secondhand fashion products through online C2C platforms, which include distancing from the system of consumption, economical motivation, and the perception of sustainability [12]. Researchers investigated how customers value fashion businesses in social media marketplaces, i.e., C2C platforms [10]. Buyer–seller discussion during sales activity shows six antecedents as perceived value, including brand availability, authenticity, origin, design, price, and perceived quality, when purchasing secondhand fashion products. The price and quality of secondhand fashion are the major things to consider for the perceived value.

### 2.4. Source of Value

Apart from the basic value structure, consumer value dimensions can be viewed according to their sources. Researchers have conceptualized customer values that are created within organizations—transaction experiences, delivered orders, and product characteristics—those related to product manufacturing and purchase [46]. These sources can be integrated into a broader framework. Five key sources of consumer value, whatever operations within and between firms make up the value chain, are proposed [47]. They are products, purchase environment, information, interactions, and possession/ownership transfer. Each of these sources may provide its own basic value perceptions, such as instrumental/functional, experiential/hedonic, symbolic, along with cost value. The sources of values are mainly divided into two parts. One is the product itself as a source of value, the other is the services along with the product as a source of value. So far, the online C2C SHF consumer perceived value is explored little from the product side while the related service areas are largely ignored.

As a source of value, products are created through activities such as market research, product development, R&D, and manufacturing. These activities provide functional/instrumental value, experiential/hedonic values (e.g., the epistemic, sensory packaging, emotional, and relational experience), symbolic/expressive value (e.g., personal attachment to the brand), sacrifice/cost value (e.g., augmented product considerations and price that reduce risk, involvement, and investment). Purchase environment is created by activities such as facility merchandising, facility design, and management. This source of value has two parts, the environment of consumption and the environment of purchase. In the consumption environment, the usage of a product enhances the perceived product value. The physical environment as the purchase environment of the store, where the product is consumed or purchased, is another value source [48]. Information, as a source of value, is created by activities associated with brand management, public relation, and advertising, such as instructions, labeling, or packaging. As another source of value, interactions between employees or systems of organizations and customers are created by activities including operations, service quality, training, and recruitment [49]. Finally, possession/ownership transfer is done by activities concerned with accounting (e.g., billing and payment), delivery, and an ownership transfer (e.g., titles, copyright, contracts).

### 2.5. Theoretical Framework

The 'basic value structure' concept restricts value perception into three categories, including utilitarian, hedonic, and symbolic. However, the 'source of value' approach keeps the value structure unrestricted and open to exploring any new emerging value. Although

there may be overlapping or interrelated value perceptions between different sources of values, the approach would be helpful to investigate a focused group of data rather than a large group of complex data and to identify any potential new or previously unidentified value concept. Therefore, the value perception found from the post-consumption of SHF is left open for exploration (Figure 1).

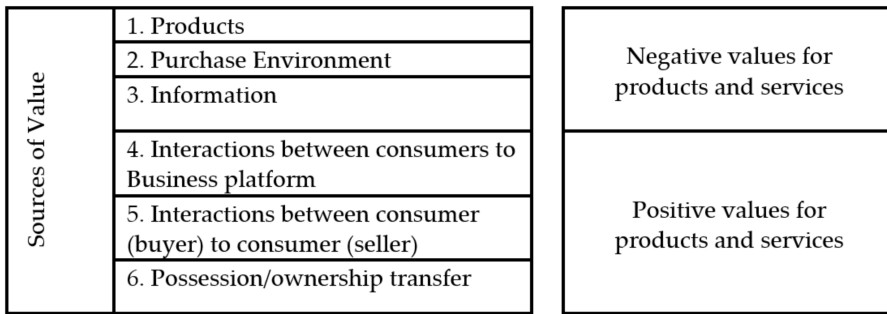

**Figure 1.** 'Source of Value' for online C2C SHF product and service resale platforms.

The theoretical framework employed in this research is adapted from the 'source of value' framework. In the online C2C market, product exchange is done on a platform where the consumer is directly or indirectly involved with another consumer who plays the role of a seller. Therefore, the source of value 'interactions' between employees or systems of organizations and consumers should be divided into 'Interactions between consumer to business platform' and 'Interactions between consumer to consumer'.

The source of value framework integrates the purchase environment and consumption environment into one source of value where purchase and consumption might be done in one place (e.g., restaurant, movie theater, etc.). However, in the C2C online environment, consumption not a part of this environment. Therefore, only a 'purchase environment' source of value is proposed. The available data were classified based on the source of value categories. The data set was further divided into positive and negative data based on sentiment.

## 3. Materials and Research Methods

The research questions are: (i) how consumers perceive the values created by the online C2C SHF business platforms" and (ii) what kind of value strategy can be adopted into mass-market SHF from luxury SHF business platforms. To address both questions, the following workflow (Figure 2) is developed. It starts with data collection followed by predictive modeling using Recurrent Neural Network (RNN) and Analysis. RNN is a type of artificial neural network. Recursive connections in its deep neural network topology let it cope with the dependence relationships in the sequential input more effectively. The process is good for working with sequential data like text. It is widely used for analyzing the syntactic structure [50].

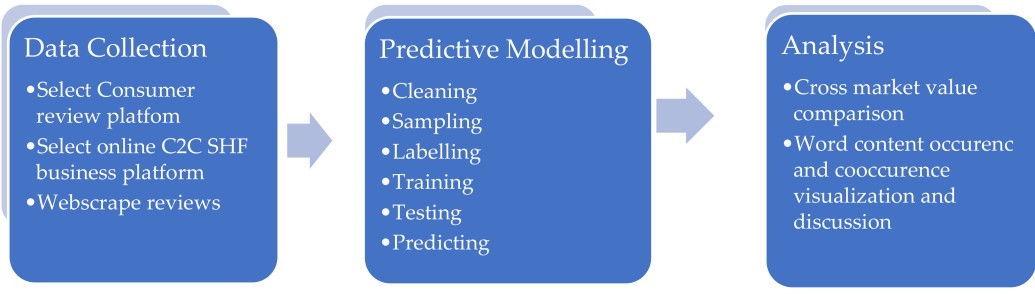

**Figure 2.** Data analysis flow chart.

### 3.1. Data Collection

Trustpilot.com was selected to collect secondhand fashion consumer post-consumption reviews. The site is widely popular as a third-party consumer review website. Most fashion companies have claimed or unclaimed reviews posted on this website. Consumers can provide their perception of using the product or service by a company on this website. They also can leave text reviews and comments along with a Likert scale rating of their overall perception, with 'one star' being the 'Bad' and 'five stars' being the 'Excellent'. Consumers are requested to rate their recent experiences that took part for the last 12 months and communicate their experience with the company [14]. Therefore, the reviews are more focused on the consumer's buying and selling experiencee than texts posted through other social media platforms, where a consumer may write anything about a company.

The top three mass-market secondhand fashion companies and the top three luxury secondhand fashion companies are selected for this research based on the annual revenue of each company. A short description of each company is given below:

The three companies for mass-market secondhand fashion are Poshmark, Thredup, and Depop. Poshmark is the top mass-market secondhand fashion product resale platform in the online media. Located in the US, Poshmark has 80 million registered accounts across Australia, Canada, and the US. Its revenue was USD 326 million in 2021. It is popular for its mass secondhand fashion product selections. Thredup is committed to helping the environment, and society, and promote good governance in its company. It is specialized in women's and kid's fashion. Its revenue was USD 251 million in 2021. Depop is another secondhand marketplace for secondhand fashion product resale. Located in the UK, it has 30 million registered users worldwide. It is highly popular in the US. Its revenue was USD 55 million in 2021.

The three luxury second fashion complaines are The Real Real, Vestiaire Collective, and Tradesy. The Real Real claims the topmost position by revenue of USD 467 million in 2021. Based the Europe, Vestiaire Collective has earned a revenue of 250-USD 500 million in 2021. The third company Tradesy exclusively saleswomen luxury fashion and claimed a share of USD 30 million revenue in 2021. Although this company allows general fashion brand products on their sites, the focus is to facilitate the resale of luxury and designer fashion brands products and make them affordable to more people. Authentication service is an added benefit provided by these companies to investigate the authenticity of luxury and designer brand products.

Consumer reviews of these companies are web-scraped from trustpilot.com using Python programming language scripts. Reviews posted from March 2015 to January 2022 were scraped. A total of 49,813 reviews were collected. Reviews rated as one or two stars were determined as negative reviews, while four and five stars were categorized into positive reviews. Reviews rated as three stars are neutral and are not considered for this research. The details of Review counts details for each selected company are given in Table 1.

### 3.2. Predictive Modelling

The review data set was pre-processed to be standardized for predictive modeling. The review texts were converted to lowercase, 'stopwords' were removed, and lemmatized–lemmatized to extract the root word. These were done using Python programming language. Out of the total reviews, 1% of the data was sampled following the stratified sampling technique. The stratification follows as every first review of each of the 100 s reviews was selected for the sample. The samples were then labeled according to the source of value criteria. A review may contain multiple sources of value based on its content. The original data set has a mixture of buyers' and sellers' reviews. For research purposes, only buyer reviews were needed. Therefore, the samples were also labeled with either buyer's or seller's reviews to help separate and extract only buyers' comments (Table 2). The labeled samples were then trained and tested using the RNN algorithm to develop models that would predict whether a review mentioned a source of value or not in each

market, namely the luxury and mass markets, and whether a review was for buyer's experience or seller's experience to help filter reviews to fit the purpose of this research. A total of 12 RNN prediction models were generated. The program code is written in Python Programming language.

**Table 1.** Data description.

| | Company | Original Review Number | Predicted Number of Buyers' Review | Number (%) of Positive Buyers' Reviews | Number (%) of Negative Buyers' Reviews |
|---|---|---|---|---|---|
| Mass Market | Poshmark | 2575 | 1782 | | 1289 (72.33%) |
| | Depop | 4470 | 2511 | | 881 (35.09%) |
| | ThredUp | 2413 | 715 | | 494 (69.09%) |
| | Subtotal | (9458) | (5008) | | 2664 (53.19%) |
| Luxury Market | The Real Real | 31,334 | 20,772 | 14,965 (72.04%) | |
| | Vestiaire Collective | 5514 | 2113 | 1315 (62.23%) | |
| | Tradesy | 3507 | 2132 | 1492 (69.98%) | |
| | Subtotal | (40,355) | (25,017) | 17,772 (71.04%) | |
| Total | | 49,813 | 30,025 | | |

**Table 2.** Sellers' and buyers' review sample.

| Seller Review Sample | Buyer Review Sample |
|---|---|
| "I started selling a few months ago and started to sell a few things but than nothing" | "As a first-time buyer, I bought a pair of trainers from the Depop site" |
| "This is not the place to send your items if you want to get a decent return on the sale" | "Bought some shoes which turned out to be a scam" |
| "Recently I sold item and the Depop team decided to take a review of" | "When I received the jacket, it was another color" |
| "Better off going to a consignment store to sale your items" | "I have not received two of my orders and all they can do is tell me to wait" |

Depending on whether it was the mass market or luxury market, 70–80% of the sample data was trained, and the rest of the data was tested. The model accuracies for the six values in the mass market were between 80.16% and 94.74%, while the numbers for the luxury market were between 86.05% and 98.84% (Table 3).

**Table 3.** RNN model accuracies for mass market and luxury market.

| Source of Value | Mass Market | Luxury Market |
|---|---|---|
| Product | 94.74% | 86.05% |
| Purchase environment | 89.47% | 93.41% |
| Information | 91.45% | 92.25% |
| Interaction between consumer to business platform | 80.16% | 94.57% |
| Interaction between consumer (Buyer) to consumer (Seller) | 94.08% | 98.84% |
| Possession/ownership transfer | 94.74% | 93.41% |

The models were then applied to the rest of the data, 99% of reviews from the mass market and luxury market were used to predict the source of value labels. For the data collected from both markets, only negatively and positively rated reviews were considered, respectively, as explained earlier. In general, mass-market SHF consumers show

negative sentiment in online review platforms [13,14]. The luxury SHF market shows the opposite [13,14]. The product–service formats for both markets are similar, however, show different phenomena. To understand the situation, only relevant reviews were collected from the third-party site. Finally, 2,664 negative reviews are found in mass-market reviews (53.19% out of all buyers' reviews), and 17,772 positive reviews are found in luxury market reviews (71.04% out of all buyers' reviews). The rest of the data are non-negative and non-positive for the mass and luxury markets, respectively.

### 3.3. Analysis

The frequencies of the source of value in different market reviews are listed in Table 4. The frequency differences between the mass and luxury markets were used to unravel consumers' ranking of values for each market. Subsequently, word co-occurrence plots were generated using Gephi software for each source of value from the two different markets. The plot visualizes the most frequently discussed words in the reviews and the most frequently co-occurred word combination in the same review. A qualitative discussion of the word frequency in the plot shows the importance of the consumer perception of a certain source of value in the C2C market platforms.

**Table 4.** Descriptive statistics of the source of value frequency.

| Source of Value | Mass-Market SHF 'Source of Value': Negative | | Luxury Market SHF 'Source of Value': Positive | |
| --- | --- | --- | --- | --- |
| | Frequency | Percentage | Frequency | Percentage |
| Products | 183 | 3.65% | 10,491 | 41.93% |
| Purchase environment | 46 | 0.92% | 1110 | 4.4% |
| Information | 595 | 11.87% | 4692 | 18.75% |
| Interaction between consumer to business platform | 1394 | 27.8% | 2004 | 8.01% |
| Interaction between consumer (Buyer) to consumer (Seller) | 274 | 5.46% | 0 | 0.0% |
| Possession/ownership transfer | 1445 | 28.82% | 9318 | 37.24% |

## 4. Findings

### 4.1. Descriptive Results

As indicated earlier, in major online consumer review platforms, overall negative sentiment has been observed among the mass-market SHF buyers and positive sentiments were found among buyers in the luxury SHF market. Thus, the descriptive statistics show that most negative values in the SHF mass market focused on 'possession/ownership transfer' and 'interaction between the consumer to the business platform'. On the other hand, in the luxury market of SHF, most of the values focused on 'possession/ownership transfer', 'product', and 'information.' However, the descriptive statistics did not show the value of 'Interaction between consumer (Buyer) to consumer (Seller)' in the luxury SHF market. Most luxury SHF platforms function through a consignment model where a consumer as a buyer has the least communication with a fellow consumer who is also a seller on that platform. While platform authority plays the role of a bridge between the buyer and seller. Table 4 presents the frequency of "source of value" for both SHF mass market and luxury market.

### 4.2. Co-Occurrence Plot Findings

The co-occurrence plots were generated for the major source of values with high frequencies for both markets. The nodes were the words and the links between two words indicated that the words co-occurred. The thickness of the link represented the frequency of co-occurrence of these words. The larger of the words indicates the more frequent they were mentioned in consumers' comments.

### 4.2.1. Source of Value: Products

The co-occurrence plot in Figure 3a illustrates that in the mass market, consumers show negative sentiment mostly through the word combinations good item, great item, great quality, great price, etc. Although consumer get great item/quality at great price, they show overall product review is negative. There are no major negative reasons seen in the word plot. This may be because of the occurrence of many minor issues like product associated 'high fee'.

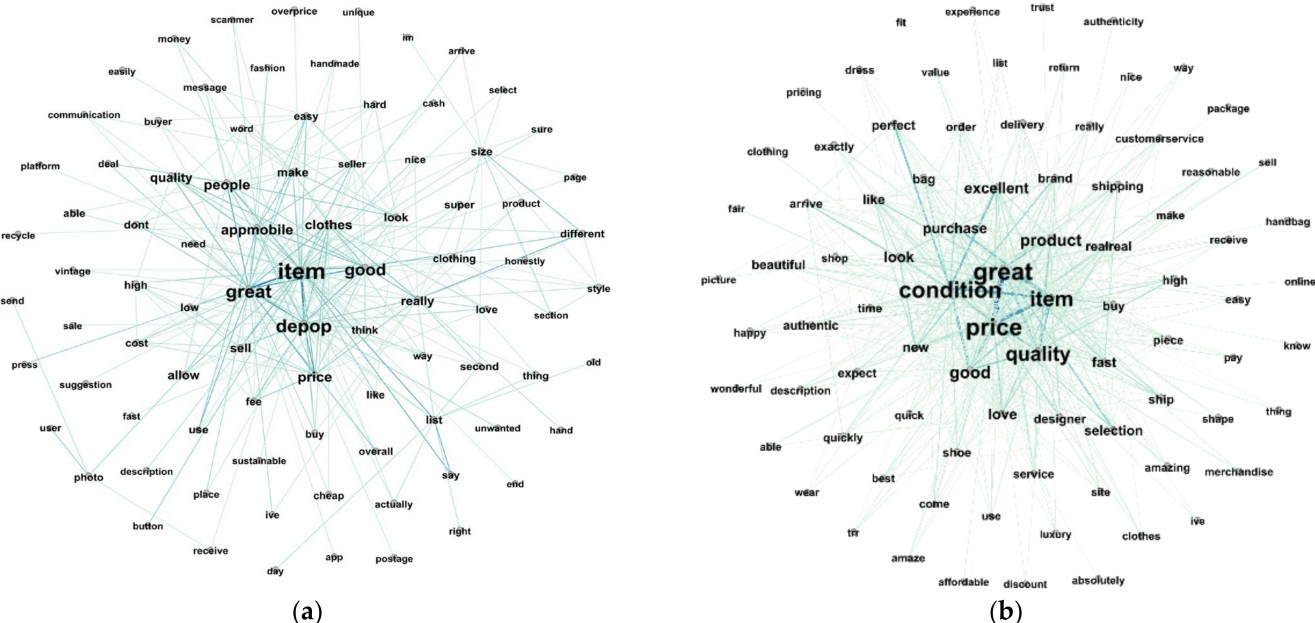

(**a**)          (**b**)

**Figure 3.** (**a**) Mass-market SHF negative reviews and (**b**) luxury market SHF positive reviews for 'Source of Value': Product.

The product values found in this study in the luxury SHF market mainly focused on the product condition, quality, and price. In the luxury SHF market, the consumer perceives positive product value when he/she receives a product in great condition at a great price. The consumer tends to relate the product quality to the product price. Such a tendency is more obvious in the secondhand market as the product is preowned and has been used. Moreover, the service of authenticity (Figure 3b) that certifies the product as authentic luxury goods adds more value to the SHF service. It should boost consumer confidence and make the SHF service more reliable.

### 4.2.2. Source of Value: Purchase Environment

The purchase environment is the least cited source of value both among the mass-market reviews and luxury SHF reviews. Although from negative reviews, the co-occurrence plot in Figure 4a illustrates few positively cited values like great app, easy app, great use, and reliable algorithm. However, consumers find various small negative sentiment on payment options, low seller rating, website environment, etc. Therefore, less frequent reviews, e.g., only 0.92% negative review, should not have major value issues.

The values found in this study in the luxury SHF market focused on easy navigation of the mobile app or website, easy usage of the app or website, and easy process of shopping (Figure 4b). This might indicate that, consumers are getting convenient shopping experience by getting an easy navigable website. Consumer's online purchase environment experience should partially resemble the in-store purchase environment to a degree where a consumer can navigate through the isles or webpages to find his desired product out of many options with minimum effort.

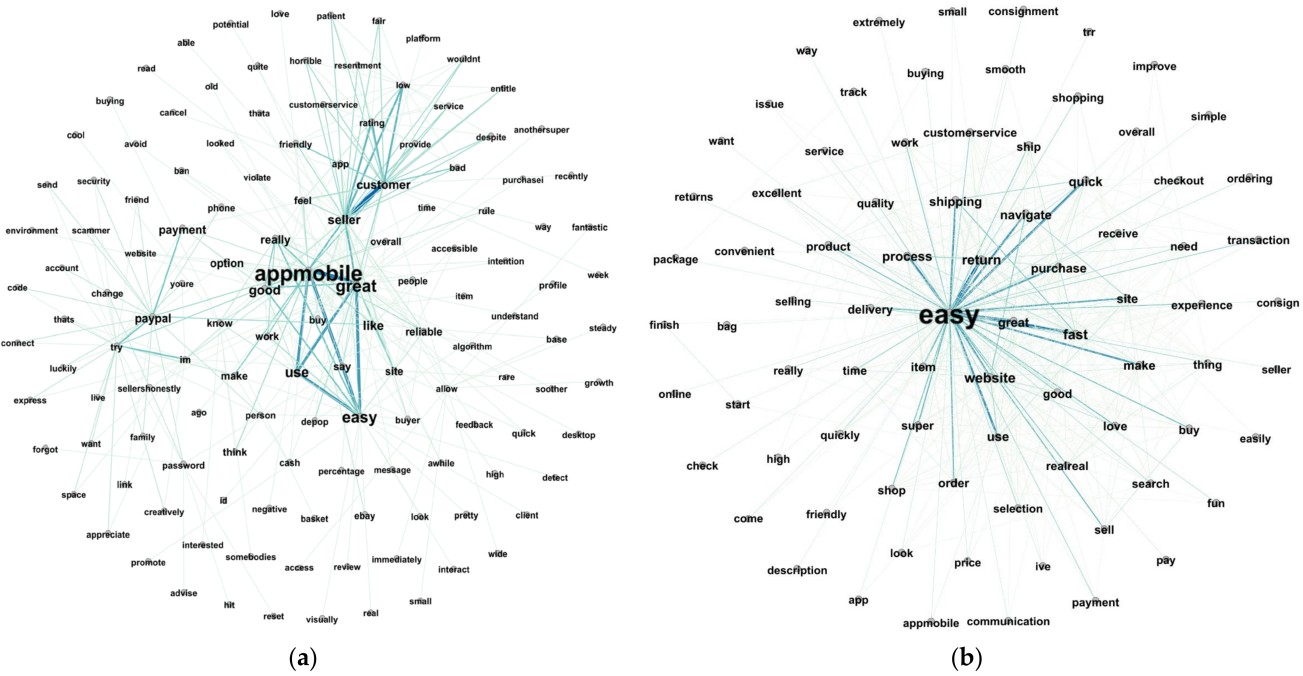

(a)　　　　　　　　(b)

**Figure 4.** (**a**) Mass-market SHF negative reviews and (**b**) luxury market SHF positive reviews for 'Source of Value': Purchase Environment.

### 4.2.3. Source of Value: Information

The co-occurrence plot in Figure 5a illustrates that in the mass market, consumers show negative sentiments mostly through the word combinations of small size, different size, cloth/jean/shoe size, correct size, size description, etc. This indicates that, there are discrepancies in product description in the site and in actual product. It may result in product return. In lot of cases, secondhand products are not returnable. The situation should frustrate the consumer and discourage him to use the service again.

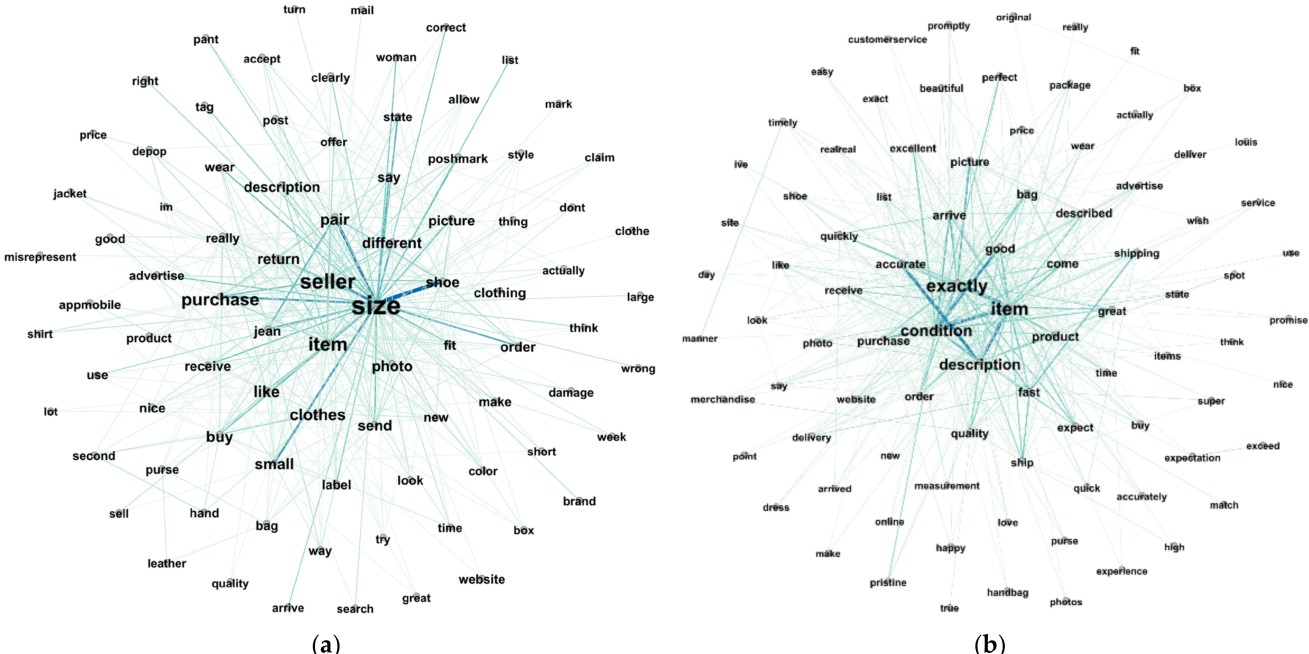

(a)　　　　　　　　(b)

**Figure 5.** (**a**) Mass-market SHF negative reviews and (**b**) luxury market SHF positive reviews for 'Source of Value': Information.

There are some positive values in which the luxury market has a competitive edge. The co-occurrence plot in Figure 5b illustrates that in the luxury market, most products received are the same as the description given for the product or in the advertisement. In some cases, the quality of used products surpasses consumer expectations generated from the given advertisement. Consumers should assess the product quality with maximum caution as the product costs a large amount of money. Therefore, for a secondhand luxury product, it is critical to depict the product with as much information as possible. Especially, one of the concerns form consumers about luxury secondhand fashion products is the authenticity and the quality of the products, providing sufficient information about the current condition and even the proof of authenticity will benefit both buyers and sellers.

4.2.4. Source of Value: Interaction between the Consumer to the Business Platform

For the values of 'interaction between the consumer to the business platform', in the mass market, the co-occurrence plot shows that the consumer as a buyer perceives negative sentiment due to not getting responses from the platform customer service (Figure 6a). In most situations, only emailing option was provided on the platform, without having any other options to contact a real customer service agent.

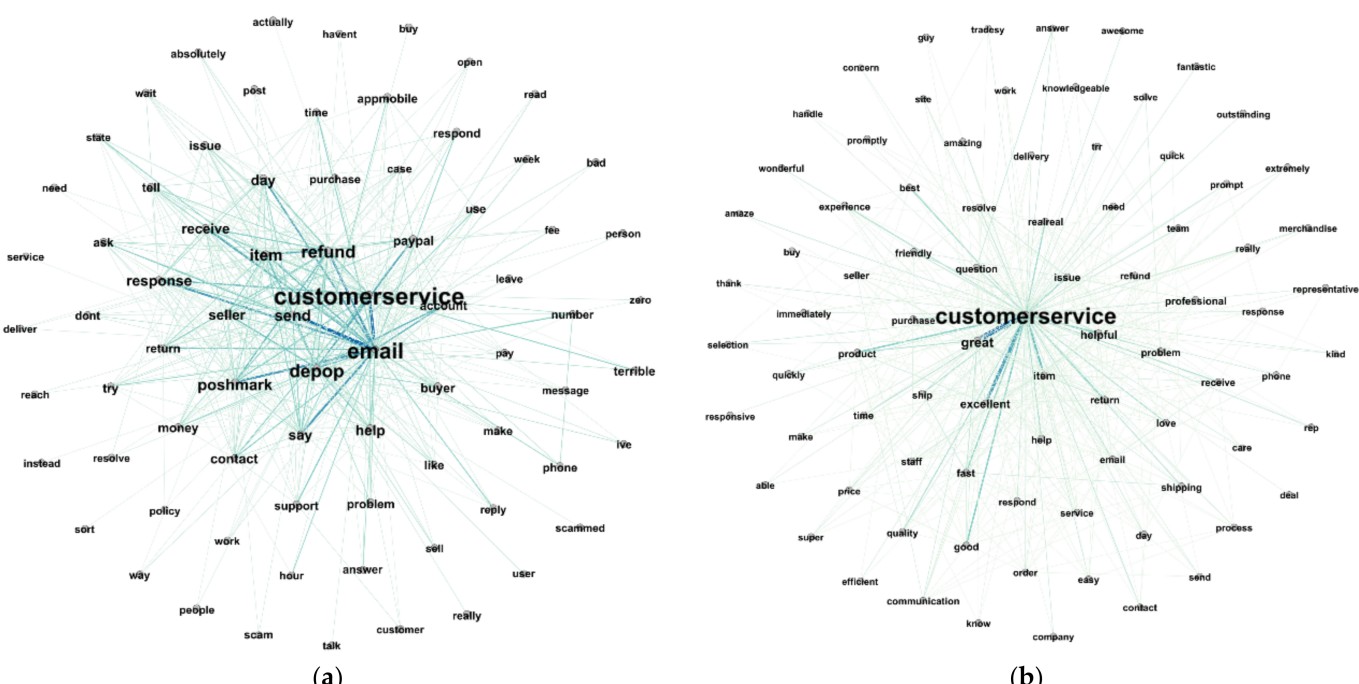

**(a)**　　　　　　　　　　　　　　　　　　**(b)**

**Figure 6.** (**a**) Mass-market SHF negative reviews and (**b**) luxury market SHF positive reviews for 'Source of Value': Interaction between consumer to business platform.

Most customer service communications were related to getting a refund, which impacted a consumer on a great level, and the company may lose the consumer due to not providing a refund in a timely manner. On the other hand, in the luxury market of SHF, customer service is central to this 'source of value' (Figure 6b). In the consignment model, professional agents of customer service work as a mediator between the buyer and the seller. As product cost increases, consumers become more sensitive to customer service handling in the post-purchase stage. Although consumers may have access to communicate with the seller directly, they still prefer to approach the platform customer service at great frequencies (Table 2). However, not getting proper treatment and their complaint not being handled properly frustrates them, and the 'source of value' was perceived as the most negative value.

### 4.2.5. Source of Value: Interaction between the Consumer to the Consumer

The source of value 'Interaction between consumer to consumer' is only cited in mass-market reviews. The co-occurrence plot (Figure 7) shows that in the mass market, consumers as a buyer show negative sentiments mostly through the word combinations seller contact, seller message, seller cancel, seller respond/response, order cancel, etc. In C2C mass-market platform a consumer can directly contact the seller. However, not getting any response or cooperation regarding the order processing may put consumer in risk followed by cancelling the order. Consumer still have the option to contact the platform authority to address the issue. However, consumer as a seller should have responsibility to entertain any issue related to the order.

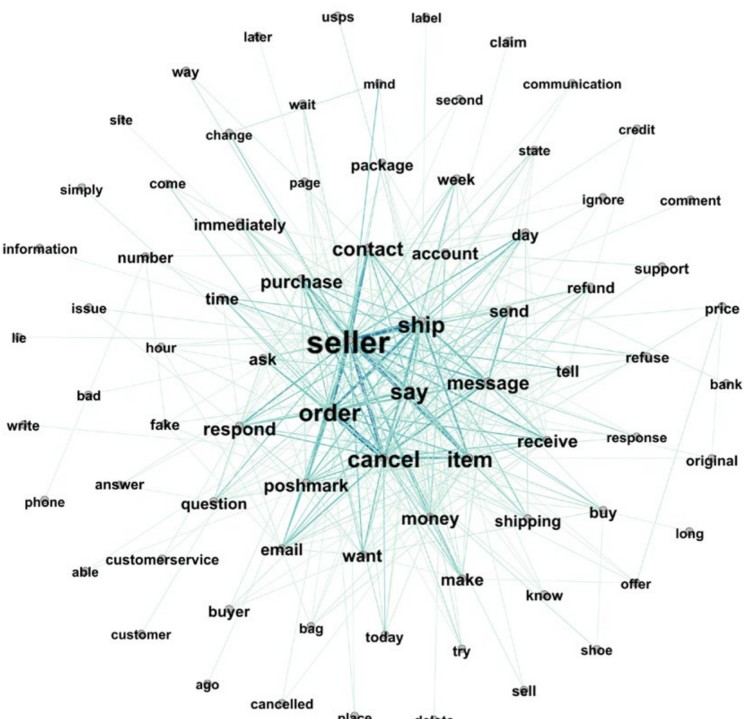

**Figure 7.** Mass-market SHF negative reviews for 'Source of Value': Interaction between the consumer to the consumer.

### 4.2.6. Source of Value: Possession/Ownership Transfer

For the possessions/ownership transfer value, the co-occurrence plot in Figure 8a shows that in the mass market, consumers had negative sentiments regarding mostly words related to return, shipping, item receiving, and refund. In the mass market, the product is shipped from a private seller who is also a consumer on that platform. The shipping process might not be standardized as no professional seller exists. In the mass market, consumers do adventure shopping to hunt for treasure and wait for the desired treasure to arrive. Delaying or never receiving the product might impact their mindset negatively toward the SHF selling service. In addition, not getting a refund for a lost or never received or returned product will add up the frustration. On the other hand, in the luxury market, consumers get a better value of fast shipping with the added benefit of quality packaging (Figure 8b). In the luxury SHF market, possession/ownership transfer service provides a convenient return policy that completes this 'source of value' to generate greater benefit than the current mass-market experience.

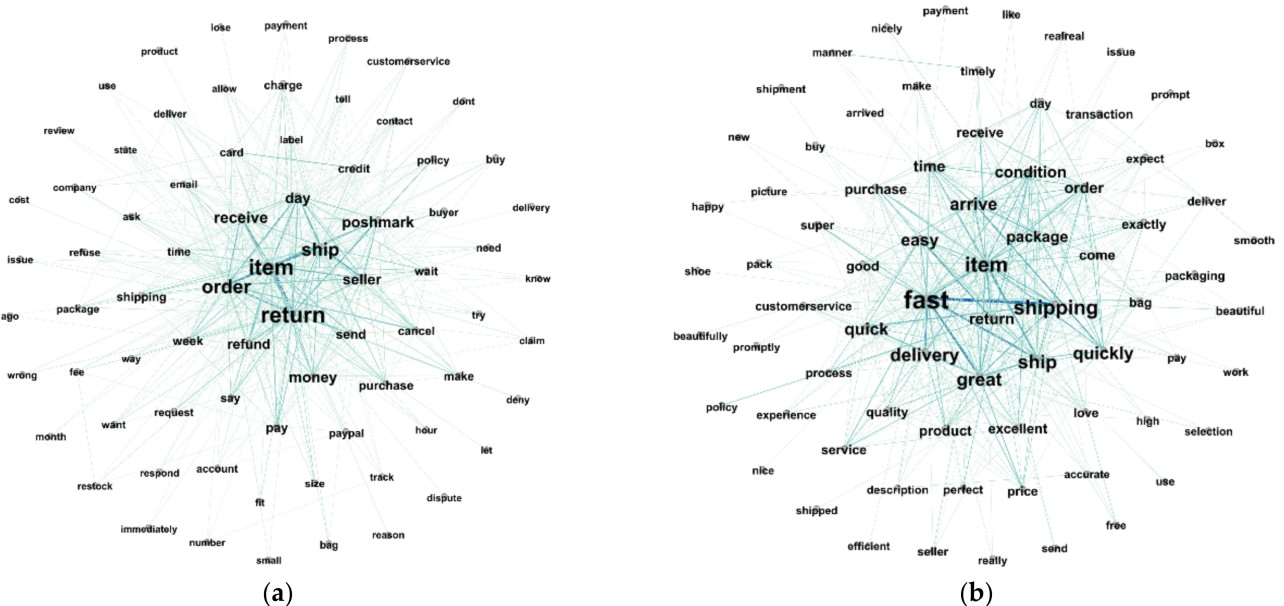

**Figure 8.** (**a**) Mass-market SHF negative reviews and (**b**) luxury market SHF positive reviews for 'Source of Value': Possession/ownership transfer.

## 5. Discussion

The findings show the potential areas of improvement in the underdeveloped sources of value area of mass-market online SHF platforms, such as possession/ownership transfer and interaction between consumer to business platform. In addition, the findings imply that ignorance of specific value areas might be crucial to reduce fashion waste. Specifically, from the cross-market analysis, it is understood that among all the 'sources of values', possession/ownership transfer-related services should be prioritized first for mass-market SHF service. The luxury market is highly improved in this single largest value area where the mass market highly ignores it. Consumer might get frustrated most when he does not receive the product on time or never receive after paying the price. The return and refund process should be smooth; the consumer is not seeing the product at first hand which may result in returning the product. Then, 'interaction between consumer to business platform' should come next. Eventually, this source of value will have a positive impact on the other source of value. Consumer wants real and fruitful communication with the platform to address an issue. Communication through email or phone with real agents should solve any service-related issues and improve consumer satisfaction in the end.

When purchasing secondhand clothing online, customers anticipate that the online retailer will send them used items that match the descriptions on their website [51]. The perceived worth of the product can be increased by employing a narrative strategy, such as outlining the background of the product [10], reducing the uncertainty and risk of using an online platform, increasing consumers' trust in the online SHF, and influence economic, social, and hedonic benefits of SHF [52]. In this research, it is seen that the mass market of SHF has not emphasized product information-related values. In practice, it is at the reseller's discretion to reveal product information in the platform. However, the platform should regulate the information criterion to maximize the amount of authentic information shared in the platform. Strict regulations on the display of product information should bring more positive sentiments to the mass market of secondhand fashion products.

Authenticating product by product experts is a differentiator between these two types of markets. Secondhand fashion consumers do not see the product before making the final purchase. As a luxury product is associated with high prices, the consumer should have concerns about the product's brand authenticity as well as quality. Obtaining an authentication certificate might give luxury SHF consumers extra motivation and encouragement to make the final purchasing decisions. However, the service is associated with

extra costs in the luxury market. In the mass market, the consumer may also be interested in the product's quality and authenticity. The mass market can implement an optimized authentication service that ensures a standard quality for secondhand products and keeps cost low. Resellers are mostly the consumer who are not expert in identifying the product brand authenticity and quality. The platform may guide and train the resellers to help identify the product brand authenticity and quality.

In C2C platforms, buyers can also be sellers. Without having professional selling experience and real customer service training, a consumer as a seller in the mass market may find it difficult to handle customer service and handle consumer complaints. This disadvantage should be compensated by the customer service provided by the platform. However, the findings from mass-market data indicate that the 'interaction between consumer to business platform' or platform-provided customer service is failed in large number. Customer service is the last resort for a consumer on online platforms. No matter the kind of secondhand merchants, keeping current clients is one of their biggest issues [53]. As the key suppliers and partners of the secondhand business, customers put a significant impact on the very existence of the secondhand store concept [53]. Therefore, providing excellent customer service is an essential aspect of the business [53]. The SHF mass market authority should increase its involvement with the consumer to facilitate the overall process (Figure 9). The current SHF model allows high interaction between the consumers and low interaction between consumers on the platform (Figure 9a) whereas the consignment model does not allow communication between consumers (Figure 9b). A more adapted model for SHF may allow high interaction among all the parties (Figure 9c). Apart from this, inspired by the luxury SHF market, the mass market may implement an optimized consignment model that increases platform involvement and does not increase the product price significantly.

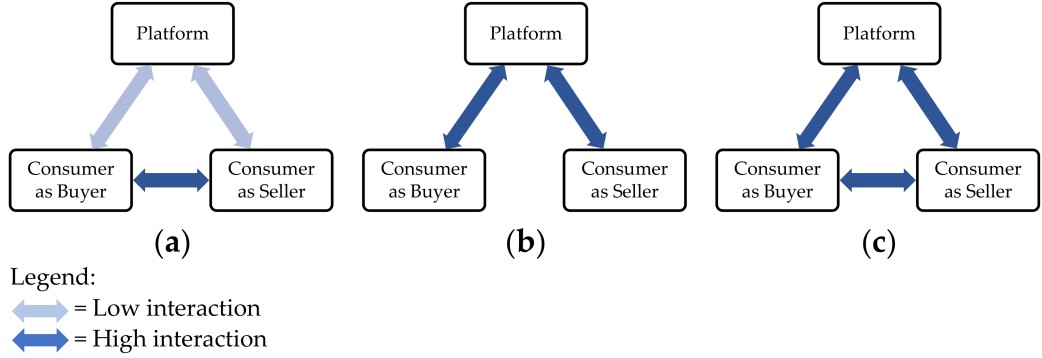

**Figure 9.** Communication practice in (**a**) Mass-market SHF, (**b**) Luxury market SHF, and (**c**) proposed mass-market SHF.

## 6. Conclusions

The purpose of this study was to evaluate consumers' perceived values of secondhand fashion (SHF) consumption in both the mass market and luxury market. To do so, a web-scraping technique was applied to download the available consumer reviews from a third-party review site. By adopting a text-mining analysis approach, this study discovered the positive motivations and negative concerns of secondhand fashion consumption from consumers' real experiences. To gain a comparative understanding, three mass-market secondhand fashion companies and three luxury secondhand fashion companies were selected for this research. Both markets sell secondhand fashion product but are oppositely perceived by the consumers. Therefore, to understand the phenomena, only negative and positive data sets of the mass market and luxury markets, respectively, were considered for this analysis. The predictive modeling technique using RNN simplified the identification of the 'source of value' of each of the reviews historically generated by the SHF consumers.

The findings show the potential areas of improvement in the underdeveloped sources of value area of online SHF platforms. Possession/ownership-transfer-related services

should be prioritized first for mass-market SHF service. The luxury market is highly improved in this area, while mass market shows the opposite trend. Interaction between consumer to business should be the second most important source of value area for mass market. The product as a source of value is the most valued area in luxury market.

To identify transferrable value strategy across the markets, the findings of this study provide contributions in both theory and practice. Mass market platforms should carefully handle the possession/ownership transfer process to improve the consumer sentiment. Moreover, they should reinforce the communication with the consumer in their platform and address any concerns in third-party media sites. Resellers should be careful to ensure they provide as much as possible information regarding the product; therefore, fellow consumer may find less discrepancies between actual product and virtual representation. The platform may guide the reseller to identify the originality and quality of the product by himself. Historical consumer perception would provide valuable insights for the future growth of the mass-market SHF. Moreover, the historic data set provides all possible instances and findings. These instances should be helpful to solve any major problems in mass-market SHF. There are some theoretical implications for the subject area literature as well. The extant literature lacks findings from secondary data readily available on the internet. Previously, there was no cross-market analysis conducted between the secondhand mass market and the luxury market for fashion products. This is the first attempt to thoroughly study consumer values of secondhand fashion in both the mass market and luxury market by using a text-mining method. The findings from this research should initiate the process of filling these gaps in the literature.

## 7. Limitations and Future Research

This study has some limitations, which might offer suggestions for further research. First of all, the research data set was collected from a single review website, which may not cover reviews from all consumers. Studying other review sites along with social media sites may give better insights. Secondly, the selected companies are the major C2C fashion platforms. Small- and medium-sized platforms should be accommodated in future research. In addition, only the consumer as a buyer's value is counted. However, in C2C platforms, the same consumer can also be a seller. Therefore, understanding the opposite role of a consumer in a C2C platform may give valuable insight to understand the overall dynamics of an SHF consumer in an SHF consumption process. In addition, follow-up studies on the comparison of related consumer behaviors might provide interesting insights. Moreover, online reviews may only reflect a portion of consumer values, and future studies applying a combination of literature reviews and consumer interviews might provide more comprehensive insights. Finally, in this study, only negative reviews from mass-market reviews, and positive reviews from luxury market reviews are considered; future research should try to find both positive motivations and negative concerns for both mass- and luxury-market secondhand fashion consumption.

**Author Contributions:** Conceptualization, H.M.R.u.H. and C.L.; methodology, H.M.R.u.H. and S.X.; validation, H.M.R.u.H., C.L. and S.X.; formal analysis, H.M.R.u.H. and S.X.; data curation, H.M.R.u.H.; writing—original draft preparation, H.M.R.u.H.; writing—review and editing, C.L.; funding acquisition, C.L. and H.M.R.u.H. All authors have read and agreed to the published version of the manuscript.

**Funding:** The APC was funded by the Louisiana State University Middleton Library.

**Institutional Review Board Statement:** Not applicable.

**Informed Consent Statement:** Not applicable.

**Data Availability Statement:** Not applicable.

**Acknowledgments:** The authors would like to thank the anonymous reviewers for their reviews and comments.

**Conflicts of Interest:** The authors declare no conflict of interest.

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
