# Peer review of "Investigating Consumer Values of Secondhand Fashion Consumption in the Mass Market vs. Luxury Market: A Text-Mining Approach"

_sustainability, doi:10.3390/su15010254_

Round 1

Reviewer 1 Report

The article is  crucial for an analysis of the SHF and so for the fashion market in general and the issues about a new model of sustainable fashion. Very original and inspiring, the findings are well presented even if a tentative of a more theoretical issue could have added a larger readers publics. A comparison with theory of consumers behaviors would have been very inspiring. Is it faisable of adding a short paragraph in this perspective in the limitations of the research? More than one reader would be more interested in looking at a possible future follow up of this research and analysis.

Author Response

Thanks for your suggestions. The discussion & conclusion was divided into two separate parts. Please see pages 15 to 16. A sentence was added in the limitation and future research section. 

Reviewer 2 Report

This study investigate consumer values of secondhand fashion consumption from online platforms in both the mass market and luxury market. Research in this area is important for promoting the sale of secondhand fashion products. However, there are some issues in the paper that need to be considered for improvement. The first is that online reviews may only reflect a portion of consumer values, and a combination of literature review and consumer interviews would have made the study more comprehensive. Secondly, the source of value is a complex issue and the research framework cited in the study should have provided more explanation as to why it was adopted. Thirdly, in the discussion and conclusion section, it would have been meaningful to have made some specific recommendations for platforms and sellers.

Author Response

Thank you for the suggestion. This study has been done and we focused on online reviews. Future research can apply a combination of literature reviews and consumer interviews to enhance the research findings. Therefore, a limitation was added on page 16.

In the 2.5 section’s first paragraph, the main reason to use this framework was reinforced. 

More specific suggestions were added, please see page 16.

Some new discussion of findings was added to address this issue in the discussion section.

Thank you for the comments. It would be helpful to specify which parts need more references
